# AN EMPIRICAL STUDY OF WEIGHTS IN DEEP CONVOLUTIONAL NEURAL NETWORKS AND ITS APPLICATION TO TRAINING CONVERGENCE

**Haihao Shen, Jiong Gong, Jianhui Li, Xiaoli Liu, and Xinan Lin**
Intel Corporation
`{haihao.shen,jiong.gong,jianhui.li,xiaoli.liu,xinan.lin}@intel.com`

## ABSTRACT

This paper presents an empirical study of weights in deep neural networks and propose a quantitative metric, *Lo*garithmical *G*eometric *M*ean of absolute weight parameter (LoGM), to evaluate the impact of weight on training convergence. We develop an automatic tool to measure LoGM and conduct extensive experiments on ImageNet with three well-known deep convolutional neural networks (CNNs). We discover two empirical observations from the experiments on same model: 1) LoGM variance is small between weight snapshots per iteration; and 2) each CNN model has a reasonable divergence region. Preliminary results show our methodology is effective with convergence problem exposure time reduction from weeks to minutes. Three known convergence issues are confirmed and one new problem is detected at early stage of feature development. To the best of our knowledge, our work is first attempt to understand the impact of weight on convergence. We believe that our methodology is general and applicable on all deep learning frameworks. The code and training snapshots will be made publicly available.

## 1 INTRODUCTION

Deep convolutional neural networks (CNNs) have demonstrated great success with break-through results on computer vision tasks such as image classification (Krizhevsky et al. (2012);Szegedy et al. (2015);Simonyan & Zisserman (2014);He et al. (2016)), object detection (Ren et al. (2015);Liu et al. (2016)), and semantic segmentation (Long et al. (2015);He et al. (2017)). With the improvement of hardware computation powers and software framework optimizations, it provides more chances for users to complete training on classical and new models. Recent work shows multi-node training with large batch size is becoming popular to accelerate the time to train significantly by leveraging more hardware resources (Goyal et al. (2017);You et al. (2017);Gitman & Ginsburg (2017)).

Regardless of the publication of CNN models, users may encounter convergence problem under their own environment. In general, convergence problem consists of two aspects: 1) training loss is not a number (NaN) or loss trend is not healthy; and 2) training cannot reach state of the art (SOTA) accuracy. To investigate the issue, users may compare the value with a reference implementation (called co-simulation). However, it cannot provide the insights on complex convergence problem with training optimization (e.g., Winograd-based convolution (Lavin & Gray (2016))) and model optimization (e.g., weight quantization (Han et al. (2015))). Without effective diagnostic tool, users have to wait and observe the training loss from time to time, debug the code or tune the hyper-parameters, and restart a new round of training. Recent research discussed the convergent learning on activation during training and proposed neuron aligns between two networks (Li et al. (2015)) to facilitate the training process. Unfortunately, there is no systematic study on convergence problem by weights, although weight snapshot is the most critical output of training.

In this paper, we propose a quantitative metric, *Lo*garithmical *G*eometry *M*ean of absolute weight parameter (LoGM). LoGM inherits from standard geometric mean but is well-tuned to support the weight snapshot with millions of learnable weight parameters trained from CNNs. We develop an automatic tool to measure LoGM by weight on top of Intel Caffe[1] and conduct extensive ex-

---

[1] https://github.com/intel/caffe

periments on ImageNet with three CNN models. We discover two empirical observations from the experiments: 1) LoGM variance is small between weight snapshots per iteration on same model; and 2) each CNN model has a reasonable divergence region. Preliminary results show our methodology is effective with convergence problem exposure time reduction from weeks to minutes. We confirm three known convergence issues and identify one new problem at early stage of feature development. To the best of our knowledge, our work is first attempt to understand the impact of weight on convergence. We believe that our methodology is general and applicable on all deep learning frameworks (e.g., TensorFlow, MXNet, and Caffe2). We recommend our metric tool is complementary to existing co-simulation tool to identify the convergence issues more effectively. The code and training snapshots will be made publicly available.

## 2 QUANTITATIVE METRIC

We define a quantitative metric LoGM in Equation (1) to measure the impact of weight. LoGM inherits from traditional geometric mean but is well-tuned to support the weight snapshot with millions of learnable weight parameters trained from CNNs. We define n is the number of weight parameters in a weight snapshot and use 10 as logarithm base to make the value more human-readable.

$$LoGM = \frac{\sum_i^n \log |w_i|}{n} \tag{1}$$

The metric indicates the similar idea of computation flow with multiplication of input activations and weight parameters from the perspective of model inference (with the assumption of negligible additions). Note that we also employ another widely-used metric standard deviation (STD) in our empirical study at the beginning. However, experimental result shows STD is not general for existing CNN models. It leads to unexpected big variance for modern CNN models with batch normalization (BN) (Ioffe & Szegedy (2015)) due to magnitude difference of weight number in convolution and BN. Therefore, it requires additional effort to handle the models with or without BN in practice. To make the metric simple and consistent, we employ LoGM as the only metric in our empirical study.

### 2.1 IMPLEMENTATION DETAILS

Algorithm 1 illustrates the pseudocode on how to compute LoGM on weight. It is straightforward with nested loops: traversing layers from a weight snapshot W at outside loop and weight parameters from a layer at inside loop. At inside loop, it accumulates the logarithm of absolute of each weight parameter. Finally, it computes the mean value of sum by the number of weight parameters.

---
**Algorithm 1** Compute LoGM by weight

---
**Input:** weight snapshot W
**Output:** LoGM
  sum = 0; num = 0
  **for** each layer L in W **do**
     **for** each parameter w in L **do**
       sum += log(abs(w))
       num += 1
  result = sum / num
  **return** result

---

We implement the algorithm with Python interface on Intel Caffe and develop the tool to measure automatically. The algorithm is general and easily applicable on all other deep learning frameworks.

## 3 EXPERIMENTS

We perform empirical study on GoogleNet-V1, VGG-16, and ResNet-50. We use the models with hyper-parameters under Intel Caffe and leverage multi-node training on Intel$^{@}$ Xeon Phi$^{TM}$ Processor 7250 with Omni-Path architecture. We employ standard ImageNet as training dataset, consisting of 1,281,167 training images and 50,000 validation images in 1,000 classes.

### 3.1 EMPIRICAL OBSERVATIONS

We summarize two empirical observations during experiments on the same well-trained model[2]. **Observation 1**: LoGM variance is small between weight snapshots per iteration. We perform the experiments with iteration level on three models and measure LoGM variance as shown in Figure 1.

Figure 1: LoGM Variance

**Observation 2**: Each CNN model has a reasonable divergence region. We extend the experiments from iteration to epoch level and show the divergence region per epoch in Table 1.

Table 1: Divergence Region

| Topology | Divergence Region |
|----------|-------------------|
| GoogleNet-V1 | (-0.09817, 0.09665) |
| VGG-16 | (-0.13456, 0.01279) |
| ResNet-50 | (-0.03719, 0.03691) |

### 3.2 APPLICATIONS

We apply the divergence region in real applications with 3 issues confirmed and 1 new detected. We demonstrate two case studies on weight quantization and Winograd convolution optimization.

#### 3.2.1 WEIGHT QUANTIZATION

Weight quantization can reduce the data size transfer on network under multi-node training by data compression and decompression. VGG-16 is a typical model with heavy weight parameters in full-connected layers. During feature development, weight partition is utilized with different scaling factor. However, there is a subtle bug in decompression with wrong scaling, which leads the model cannot reach SOTA accuracy. We measure the divergence variance on the first two weight snapshots per iteration and find that the value is out of the reasonable region. After bug fixing, weight quantization works well. Comparing with regular training by weeks on single CPU node, the case shows our methodology can shorten the debugging cycle significantly from weeks to minutes.

#### 3.2.2 WINOGRAD-BASED CONVOLUTION OPTIMIZATION

Winograd is a new class of fast algorithms for convolutional neural networks to speed up convolution computation on small filters. Intel[@] math kernel library for deep neural networks (MKL-DNN) is an open source performance library for deep learning applications intended for acceleration of deep learning frameworks on Intel architecture[3]. We enable Winograd convolution algorithm on Intel Caffe with MKL-DNN Winograd primitive and perform the training on VGG-16. The divergence variance is 0.16922 on first two iterations in VGG-16 training, which is out of valid divergence region. We report the convergence issue and confirm with MKL-DNN team.

### 3.3 SUMMARY

We believe the above observations are not limited to CNN models used in our experiments. We recommend users measure reasonable divergence region on their own CNN model as baseline and apply the similar idea in real applications.

---

[2]Well-trained model is proved to reach SOTA accuracy.
[3]https://github.com/01org/mkl-dnn

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
