# OpenReview forum: "An Empirical Study of Weights in Deep Convolutional Neural Networks and Its Application to Training Convergence"
_ICLR.cc/2018/Workshop — Reject_

### Official Review · AnonReviewer3 · 2018-02-27
**Incremental impact, experiments are unclear**

**Rating:** 3
**Confidence:** 4

**Review:**

The paper suggests to use log geomentric mean as debugging tool when training DNNs. However I have several major concerns on it:
1. The idea of using geomentric mean of the weights is quite straightforward and doubtfully can serve a topic for the research paper.
2. The exeriments description is very unclear. It is obvious how the geomentric mean can by used for debugging. The authors claim that the geometric mean should vary within narrow interval otherwise something is wrong. However no experiemntal evidence is provided.
3. The range of log geometric range seems strange. It contains zero in all three cases hence the geometric mean for the weights is abuot one which is weird.

I think the impact of the paper is quite low and the experimental description is insufficient.

---

### Official Review · AnonReviewer2 · 2018-03-09
**The paper is quite confusing.**

**Rating:** 4
**Confidence:** 4

**Review:**

I like the idea to study the weights in deep neural networks. But the intuition of this paper is very unclear. I think the author could remove section 2.1 to put more experimental results. The observations are not clear. For example,observation 1 claims that "LoGM variance is small between snapshots per iteration", which is hard to understand. And the author should define the quantity of "small" by showing the normal range of LoGM in the whole training process. Observation 2 needs to properly define the "divergence region". In addition, conclusions drawn by having three networks trained on a single dataset seems unconvincing.

---

### Official Review · AnonReviewer1 · 2018-03-09
**Quantitative metric to detect convergence problems in CNN training**

**Rating:** 5
**Confidence:** 5

**Review:**

This paper proposes a metric called LoGM to evaluate the evolution of the CNN weight during the training. The authors claim that LoGM can be used to detect convergence problems in the CNN training.

The paper is confusing and hard to understand.  There are different aspects that are unclear. For example:

- Authors state that "the problems in the convergence of CNN training can be due to obtining 'not a number' in the evaluation of the loss function, or due to the fact of non obtaining state-of-the-art accuracy". I think this is an unfounded simplification. For example, the problem could be also the batch size, among others.

- LoGM is defined as the average log of the CNN weigths. What is the intuitive idea behind this measure? Algorithm 1 is unnecessary.

- Figure 1 is not properly discussed. In my opinion these results (and also the results of Table 1) do not show the potential of the metric.

- In section 3.2. states "We apply the divrgence region in real applications with 3 issues confirmed and 1 new detected". I do not know what this means in the context of the paper.

- Authors mention often the concept "reasonable divergence region", but the term is not properly defined or discussed.

I think this paper is not clear. The work pretends to make an observation but, from my viewpoint, it is not solid enough to be published.

---

### Decision · Program_Chairs · 2018-03-20
**ICLR 2018 Workshop Acceptance Decision**

**Decision:**

Reject

**Comment:**

Based on the reviews, this paper has not been accepted for presentation at the ICLR workshop. However, the conversation and updates can continue to appear here on OpenReview.